# Analysis and Sequence Alignment of Peste des Petits Ruminants Virus ChinaSX2020

**DOI:** 10.3390/vetsci8110285

**Published:** 2021-11-22

**Authors:** Lingxia Li, Jinyan Wu, Xiaoan Cao, Jijun He, Xiangtao Liu, Youjun Shang

**Affiliations:** State Key Laboratory of Veterinary Etiological Biology, Lanzhou Veterinary Research Institute, Chinese Academy of Agricultural Sciences, Lanzhou 730046, China; lilingxia963@foxmail.com (L.L.); wujinyan@caas.cn (J.W.); caoxiaoan@caas.cn (X.C.); hejijun@caas.cn (J.H.); liuxiangtao@caas.cn (X.L.)

**Keywords:** PPRV, genome, ChinaSX2020, sequence alignment

## Abstract

The peste des petits ruminants virus (PPRV) mainly infects goats and sheep and causes a highly contagious disease, PPR. Recently, a PPRV strain named ChinaSX2020 was isolated and confirmed following an indirect immunofluorescence assay and PCR using PPRV-specific antibody and primers, respectively. A sequencing of the ChinaSX2020 strain showed a genome length of 15,954 nucleotides. A phylogenetic tree analysis showed that the ChinaSX2020 genome was classified into lineage IV of the PRRV genotypes. The genome of the ChinaSX2020 strain was found to be closely related to PPRVs isolated in China between 2013 and 2014. These findings revealed that not a variety of PRRVs but similar PPRVs were continuously spreading and causing sporadic outbreaks in China.

## 1. Introduction

Peste des petits ruminants (PPR), caused by the PPR virus (PPRV), is an acute and fatal contagious disease affecting the majority of small ruminants. It mainly affects the sheep industry worldwide, and the range of hosts infected with PPRV continues to increase [1,2]. PPRV is a member of the genus *Morbillivirus* in the family *Paramyxoviridae*. It is a negative-strand RNA virus, and the viral RNA of PPRV is 15.9 kb in length. According to the N and F genes of PPRV, the phylogenetic tree is divided into four lineages (I–IV) [3]. However, lineage IV is the most commonly found epidemic strain in China and other Asian regions.

In the last 20 years, PPRV has posed a great threat to the sheep and goat industry, as well as to public health. The spread of PPRV continues to increase among unvaccinated domestic small ruminants. The research has shown, through serological investigations in African countries and where PPRV is endemic, that PPRV repeatedly infects various wild animals [4]. Globally, PPR epidemics still have regional epidemic and multiregional distributions. Since PPR was first reported, this disease has continued to spread in more than 70 countries throughout the world [5]. In our previous study, we found that PPRV was circulating among wild goats in the Qilian, Helan and Yinshan mountains of China [6]. In addition, the geographic range of PPRV infection continues to expand, reaching previously uninfected areas. Now, the number of infected counties continues to increase, extending into Central and East Asia and Europe [7].

This virus spreads from livestock in multiple locations and at different times. The host range of PPRV infection is gradually expanding, with research showing that PPRV has the potential to adapt to a variety of new hosts [8,9]. According to the reported data from January 2014 to June 2018, PPR occurred every year in Hunan province of China. The latest outbreak of PPR was reported in Hunan province on 15 June 2018, with no new outbreaks ever since. In addition, Anhui, Jiangsu and Yunnan provinces had 30, 33 and 56 epidemic locations, respectively [10,11]. These studies suggested that the cross-border transmissions by wild and domestic animals were closely related to the spread of PPRV [12]. Every year, PPR infections cause huge economic losses worldwide. At present, there is no effective treatment or specific medicine for PPR, and prevention or control is mainly carried out through vaccine immunization. Therefore, a novel vaccine is a promising tool to help control this disease [13]. However, regional epidemics are still frequent due to immunization failures or other human factors. Thus, the continuous surveillance and monitoring of the circulating strains of PPRV would make a major contribution to the global campaign to eliminate this virus [14,15,16]. The objective of this study is to evaluate one whole-genome sequence of PPRV collected from Shannxi province of China, in order to provide primary data for epidemiological analysis of PPRV in China, even around the world.

## 2. Materials and Methods

### 2.1. Cells and Virus

Vero cells were cultured with MEM medium containing 10% FBS and 100 μg/mL of streptomycin and 100 IU/mL of penicillin. Goat tracheal epithelium cells (GTC) were generously provided by Prof. Chu Yuefeng (Lanzhou Veterinary Research Institute) and were cultured in RPMI 1640 Medium (Gibco, Grand Island, NY, USA) containing 10% FBS (Hyclone, Logan, UT, USA), 100 μg/mL of streptomycin and 100 IU/mL of penicillin. Then, the cells grew after five times for passages and were then stored in our laboratory. For virus infection, the experiments were performed similarly as in [6].

### 2.2. Indirect Immunofluorescence Assay

GTC were infected with PPRV ChinaSX2020 at the indicated multiplicity of infection (MOI) of 2.0 at 37 °C for 2 h. The PPRV inoculum was removed, and fresh 1640 medium containing 2% FBS was added back onto cells. RPMI-1640-treated cells were set as mock. PPRV infection with GTC at 24 h, 48 h, and 72 h were collected. Then, the cells were fixed with 4% paraformaldehyde at RT for 30 min. Cells were blocked with PBS containing 0.05% Tween 20 and 2% skim milk powder for 1 h at 37 °C. PPRV-N polyclonal antibody (1:200) was added and incubated at 4 °C overnight. The cells were washed three times with PBS and then incubated with FITC-conjugated secondary antibody (1:200) for 1 h at 37 °C. After washing five times with PBS in a dark place, then cells were detected under a fluorescence microscope.

### 2.3. RNA Extraction, Quantitative Real-Time PCR and Genome Sequencing

Clinical samples including spleen, small intestine, lung, and mesenteric lymph nodes were ground with PBS. For RNA extraction, the cell supernatant was discarded, the 500 μL Trizol was added. Then, total RNA was extracted and was reversed transcription as cDNA using a GoScript^TM^ RT reagent Kit (Promega, Madison, WI, USA). Reverse transcription was performed at 42 °C for 15 min, and 72 °C for 15 min, in a reaction mix containing 2 μg of isolate RNA, 1 × Reverse Transcription Buffer, 2.5 mM MgCl_2_, 1 mM each dNTP, 1 U/μL Recombinant RNasin^®^ Ribonuclease Inhibitor, 15 U/μg AMV Reverse Transcriptase and 0.5 μg Oligo(dT)_15_ Primer or Random Primers in a final volume of 20 μL [17].

To detect PPRV infection with GTC, cells were infected with PPRV ChinaSX2020 for 24 h, 48 h and 72 h. Quantitative real-time PCR (qPCR) was performed using PPRV-H primers (forward: 5′ -CTGAATACCAACATTGAG-3′, reverse: 5′ -GAGGAACTTAATCTTATCG-3’), and goat β-actin primers (forward: 5’-ACCAAACAAAGTTGGGTAAGG-3′, reverse: 5′-AGTCCACATCGCTGTCGTCAGATC-3′) Then, detections were carried out in the Mx3005p system (Agilent Technologies, Palo Alto, CA, USA) following the manufacturer’s instructions. Data analysis was performed using the 2^−∆∆CT^ relative quantification method. For PPRV detection by PCR, N gene (351 bp) was used, and the primers were from reference [18].

For PCR amplification, 11 pairs of primers were designed to target the reference sequences of PPRV. The full-genome sequence of PPRV was spliced by a nested PCR assay. The amplified products were separated on a 1.0% agarose gel and purified with a DNA gel extraction kit. Then, the PCR products were sequenced using standard Sanger methods at TsingKe Biological Technology (Xi’an, China). The PCR primers designed to amplify the 11 overlapping fragments are listed in Table 1.

### 2.4. Serology for Detection of Antibodies Directed against PPR Virus

For the last three years, our laboratory has been monitoring PPRV infection and has collected a large number of sheep serum. Then, samples of sheep were collected for preliminary PPR serological analysis in the Lanzhou Veterinary Research Institute, Chinese Academy of Agricultural Sciences (LVRI-CAAS, Lanzhou, China). Serological tests were performed using blocking ELISA (bELISA) based on the PPRV N protein purchased from Qingdao ReGen diagnostics development center.

### 2.5. Phylogenetic Analysis

The genome sequences of PPRV obtained in this study have been deposited in GenBank. The phylogenetic relationship of PPRV strains was analysed using MEGA6.0. Additionally, the phylogenetic tree was further constructed using the neighbour-joining method by bootstrap analysis with 1000 repetitions.

## 3. Results

### 3.1. Serological Analysis

To investigate the prevalence of PPRV in China, a retrospective serological study was performed on the sera samples collected from some provinces in China during 2018–2020. We found that 75 out of a total of 573 (13.09%) were positive, while 413 (72.07%) were negative and 85 (14.83%) were inconclusive when tested using the N-based PPR-bELISA according to the instructions (Table 2). These results revealed that PPRV might be epidemic in some areas of China.

### 3.2. PPRV Identification

To further understand the epidemiological surveillance of PPRV, in July 2020, clinical samples were collected from dead domestic milk goats, which were suspected to have died of PPRV in a sheep yard of Xi’an country Shannxi province. Prior to their death, they displayed symptoms of PPR infection, such as cough, dyspnoea, mucopurulent ocular, nasal discharge and diarrhoea [19]. A complete necropsy was performed, and tissue samples including lung, spleen, small intestine and mesenteric lymph nodes were sent to the laboratory on ice for further detection. These tissues were ground with cold PBS. Then, they were centrifuged at 4 °C, 8000 rpm for 30 min and were stored at −80 °C for further investigation. qPCR analysis investigated that the spleen, small intestine, lung, and mesenteric lymph nodes were positive for PPRV (Figure 1A). Moreover, PCR results showed that the N gene amplification size was 351 bp (Figure 1B). Taken together, these data suggested that the domestic milk goats were infected with PPRV.

Afterwards, the ground and homogenized spleen was used for virus infection. The results showed that GTC produced more cytopathic effect (CPE) at 48 h and 72 h during PPRV infection (Figure 2A). Additionally, an indirect immunofluorescence assay (IFA) indicated that the expression of PPRV-N protein was obviously observed during PPRV infection (Figure 2B). And the mRNA level (Figure 2C) and virus titer (Figure 2D) of PPRV increased significantly after infection, indicating that PPRV was replicated in GTC cells.

### 3.3. PPRV Identification and Sequencing

Additionally, the virus was concentrated and purified. Transmission electron microscopy revealed spherical, enveloped virus particles, with a mean diameter of 250 nm (range—200–400 nm) (Figure 3A), which was consistent with those of the reported PPRV particles. This PPRV strain was named ChinaSX2020. 11 fragments were amplified by PCR using ChinaSX2020 cDNA (Figure 3B), whereafter the amplifications of 11 segments were sequenced and spliced to obtain the complete genome sequence of ChinaSX2020.

### 3.4. Multiple Alignment and Phylogenetic Analysis

The complete genome sequence obtained from this study has been submitted to GenBank and the accession number was MW344288. Afterwards, the phylogenetic analysis tree of PPRV was constructed. We found that the ChinaSX2020 strain belonged to the lineage IV genotype, which is the same as the ChinaGS2018 isolated in our laboratory in 2018. These two sequences showed good conformity with the PPRV complete genome sequences in China available on GenBank (Figure 4). The genome of ChinaSX2020 is 15,954 nucleotides long, with a GC content of 48.18%. Compared with the full genomes of other PPRV strains, it was found that the genome of ChinaSX2020 showed the highest nucleotide sequence identity to PPRV isolated in China between 2013 and 2014.

## 4. Discussion

Outbreaks of PPRV in free-ranging, wild artiodactyls can not only decimate them, but even threaten wildlife populations and ecosystems [20]. Over recent years, PPRV has been spreading across Asia and Africa [21]. There are more than 70 million sheep and goats, according to the official data published by the OIE in China. Additionally, the incidence rate of PPRV in Asia and Africa is high [22], which will have a major impact on the region’s breeding industry. Due to the potential effects of the viral synonymous codon usage bias exhibited by an RNA virus, PPRV is continuously evolving [23]. In July 2007, the lineage IV PPRV was firstly descripted according to the reported PPR outbreaks that have occurred regularly from November 2013 until now in different regions of China [24]. Lineage IV was also found across the Middle East, Southern Asia and several African territories [25]. Additionally, with the increasing population densities of goats, the probability of PPRV occurrence was still high, which also threatens food security. Therefore, achieving global PPRV eradication appears to be particularly important [26].

In this study, the positive sera from these surveyed samples may be a result of anti-PPRV antibodies’ presence, the spread of PPRV in Asia, weak PPRV immunization measures, the existence of hot and humid climate conditions or other human factors [22]. Afterwards, the ChinaSX2020 was spliced by eleven fragments, and missing short sequences were usually amplified with a new set of primers. Polymerase chain reaction (RACE PCR) was used to rapidly amplify the cDNA ends to obtain the genome extremities [27]. ChinaSX2020 had the same length as most of the other PPRV genomes sequenced in China. ChinaSX2020 did not show sequence diversity like ChinaGS2018, but they had high homology, which suggested an ongoing circulation of PPRV in China. These findings showed that the recent PRRV is very similar to that isolated during the PRR outbreak in China between 2013 and 2014. These findings revealed that not the variety of PRRVs but the similar PPRVs were continuously spreading and causing sporadic outbreaks in China. In the absence of further information, it may originate from the PPR outbreaks in China during 2013–2014.

Over the last several years, PPR outbreaks have had a devastating effect on the economy over the last several years in China. Our previous study speculated that transboundary transmission may be an additional factor influencing the high prevalence of PPRV mortality among goats and sheep [6]. Although the majority of small ruminants are specific hosts to PPRV, domesticated cattle and buffalo are described as dead-end hosts [28]. Livestock production is associated with PPRV, such as the free movement of small ruminants, which could lead to the spread of PPRV. At present, efficacious vaccines are available against PPRV, such as Nigeria75/1 and Sungri96/1, but poor disease surveillance, low vaccine coverage and uncontrolled animal movements, exacerbated the spread of PPRV across the world [29]. There is a high risk of importing small ruminants from abroad, especially from countries with large populations of sheep and goats, which are likely to carry PPRV [30]. Additionally, vigorous monitoring is valuable for controlling and eradicating PPR [31].

To sum up, this study reported a new strain of PPRV named ChinaSX2020, which was from a domestic milk goat from Shannxi province in China. This strain was sequenced and identified as belonging to lineage PPRV IV. Additionally, the genome of ChinaSX2020 showed a 99% nucleotide sequence identical with other PPRV in China between 2013 and 2014. This report will provide new information regarding the prevention and epidemiological surveillance of PPRV to eradicate the PPR disease globally.

## Figures and Tables

**Figure 1 vetsci-08-00285-f001:**
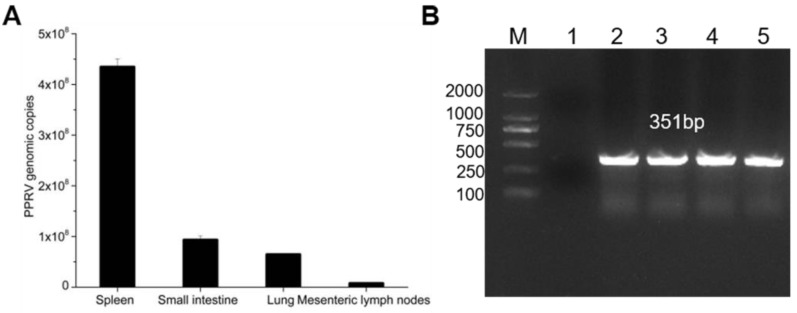
Identification of the PPRV. (**A**) PPRV infection was detected by qPCR. Genomic copies of PPRV per gram of tissues; (**B**) PPRV infection was detected by PCR with N fragment. M: DL2000. (1) Negative control; (2) lung; (3) spleen; (4) small intestine; (5) mesenteric lymph nodes.

**Figure 2 vetsci-08-00285-f002:**
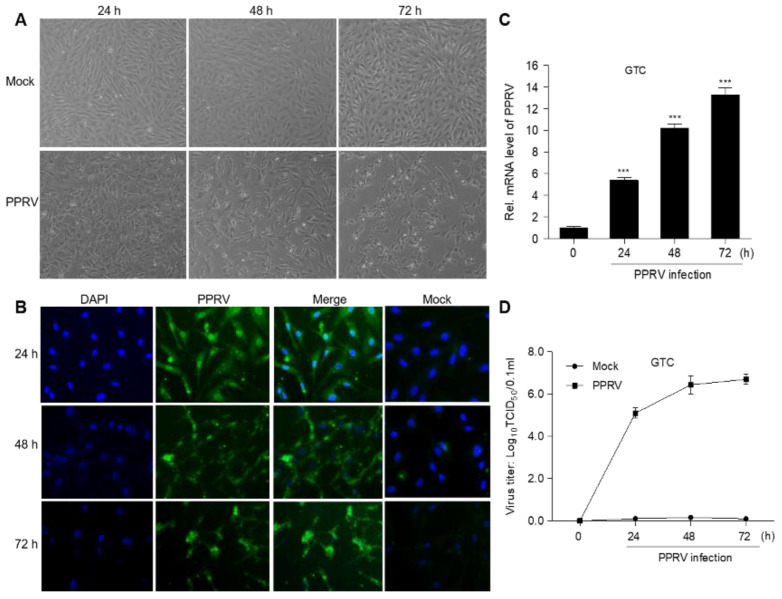
PPRV infected with GTC. (**A**) CPE was observed by microscope; (**B**) indirect immunofluorescence of GTC infected with PPRV; (**C**) PPRV replication was detected by qPCR; (**D**) virus titer of PPRV was determined by qPCR. The data represent the mean ± SD of three independent experiments. One-way ANOVA; *** *p* < 0.001.

**Figure 3 vetsci-08-00285-f003:**
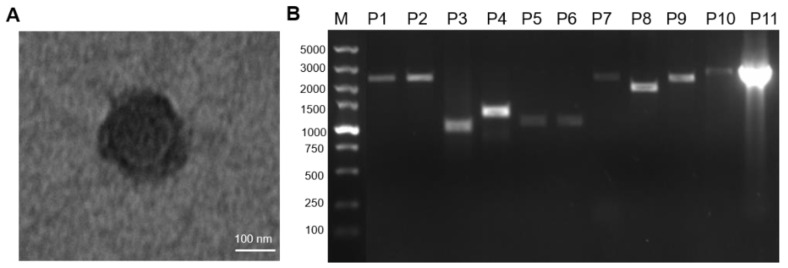
Electron microscopy image of virus particles and amplification of the full-length genome of PPRV. (**A**) Electron microscopy image of PPRV particles; (**B**) the full-length genome of PPRV was amplified by 11 fragments.

**Figure 4 vetsci-08-00285-f004:**
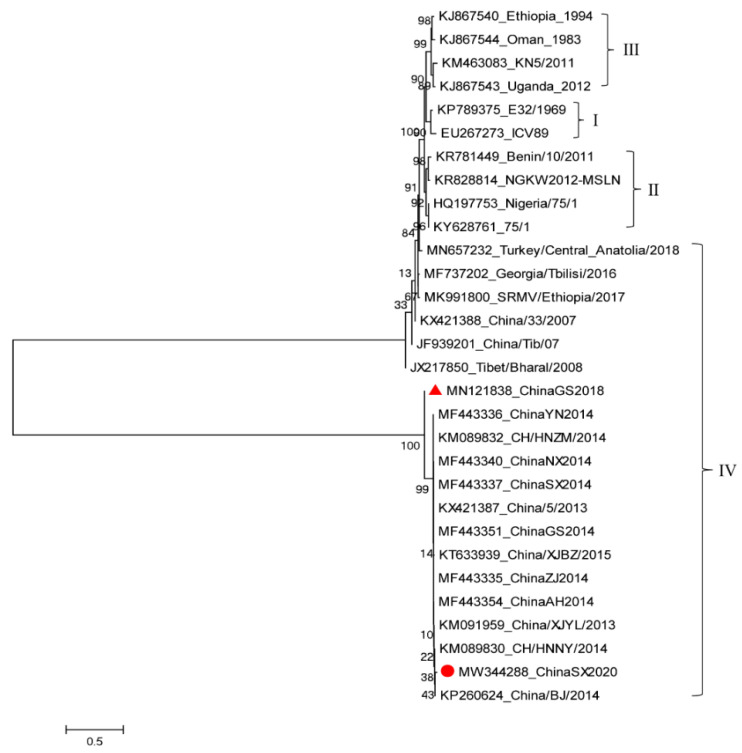
The phylogenetic tree analysis of PPRV. The ChinaSX2020 sequenced in this study is marked as a red dot in the tree. The ChinaGS2018 sequenced in our previous study is marked as a red triangle in the tree.

**Table 1 vetsci-08-00285-t001:** Primers used in this study.

Gene ID	Forward Primer (5′-3′)	Reverse Primer (5′-3′)
P1	ACCAAACAAAGTTGGGTAAGG	ACCAAACAAAGTTGGGTAAGG
P2	GATTGAAGGACTCGAGGATGCTGAC	TGATGATGACATCATCGTAGACACGG
P3	ACCCTAGAAGATACATAGTCGGCTCATG	TCTCGTATGGACTTGGCCCCTAA
P4	GGACGCAGAAAGGAAGGAGACAC	CCCCCTGAAACATTCCTGAAGCA
P5	GCCAAGCCACCAGACTCTGGTTATA	CATGTCTGTGTGTGATGCCAGATGA
P6	GCACCAATTTAGGCAATGCAGTCAC	CCCGAGAGTCAAAGATTGCAGCTTT
P7	TACAAGGCTGCGGTCAAGTCAATTG	ATATCTCTGGTCTATGGCCATGGCT
P8	TCGCGAGACCTCGTTGTGATAATTG	GCCGCTCTGGTTTCATCCACTATAG
P9	GCTGCACTGAAGAATGAGTGGGATTC	AGAGGTTCTCAAGGATCCCAAGACC
P10	GACAATCAGACAATCGCAGTGACGA	TGGATGTGGAGACTGGAGTGATCAT
P11	GACATCCCTTGTGAGGGTTGCAAGATAC	ACCAGACAAAGCTGGGAATAGATAC

**Table 2 vetsci-08-00285-t002:** Retroactive serological survey for the detection of anti-PPRV antibodies in sera collected in China.

				NPPR-bELISA Test Results		
	Time of Sample		No. of Samples	Positive	Negative	Inconclusive
County	Collection	Sample Type	Analysed	PI ≥ 60%	PI ≤ 40%	PI: 40–60%
Inner Mongolia	Mar. 2018	Serum	80	14	56	10
Jinchang	May. 2018	Serum	192	15	142	35
Zhang Jiachuan	Oct. 2018	Serum	10	2	8	0
Longnan	Jun. 2019	Serum	90	12	62	16
Jinchang	Aug. 2019	Serum	145	25	98	22
Shaanxi	Jul. 2020	Serum	56	7	47	2
Total			573	75/573 (13.09%)	413/573 (72.07%)	85/563 (14.83%)

The N-based PPR-bELISA test results are interpreted as follows: PI values below or equal to 40% (PI ≤ 40) are negative, PI values greater than 60% (PI ≥ 60) are positive, and PI values greater than 40% and below 60% are doubtful.

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
