# Peer review of "Analysis and Sequence Alignment of Peste des Petits Ruminants Virus ChinaSX2020"

_vetsci, 2021, doi:10.3390/vetsci8110285_

Round 1

Reviewer 1 Report

The manuscript “ Analysis and Sequence Alignment of Peste des Petits Ruminants (PPR) Virus ChinaSX2020” documents isolation of a field strain of PPRV collected from a domestic milk goat. The authors describes how the isolate was obtained, and confirmed its etiology by different assays. 

  1. The manuscript needs some minor editing for proper English and style. 
  2. The abstract is disorganized and redundant. Identification of the virus before talking about RT-PCR?  
  3. Line 157: the CT value does not bring anything to the results if there is no standard the reader can refer to.
  4. Figure 1 A: the number of genomic copies are mentioned without mentioning the standard used. Is it per gram of tissues?
  5. Name of statistical test used for Figure 2C analysis is missing.
  6. it says that the data are normalized to an housekeeping gene. However, line 95 authors say stat analysis was performed using the delta delta method, which has nothing to do with how the data should be analyzed. Was it actually done?
  7. Table 2: What was the origin of the sample giving the 100% reference value (negative control serum)? 
  8. ChinaSX2020 is not plotted in the tree. There is a mislabeling. 

Author Response

Thank you very much for your letter regarding our manuscript entitled “Analysis and Sequence Alignment of Peste Des Petits Ruminants Virus ChinaSX2020” (Manuscript Number: vetsci-1441706).

We also thank the anonymous reviewers for providing their comments and suggestions that are helpful for improving our manuscript. Based on their requests, we have carefully evaluated their comments and suggestions, responded point-by-point and revised the manuscript accordingly. Our point-by-point responses are listed below this letter.

The language has been also modified by native English speaking. We missed two authors when we submitted, and we have added them this time.

we have addressed all the comments from two reviewers. The files include:

  1. A point-by-point response to the reviewer comments (file: Responses to reviewers R1).
  2. The revised manuscript, all changes are highlighted in green (file: vetsci-1441706-R1).

We hope that the revised manuscript has satisfactorily addressed all the concerns raised by the reviewers.

Looking forward to hearing from you soon.

With kind regards,

Yours sincerely,

Lingxia Li & Youjun Shang

Reviewer #1

The manuscript “Analysis and Sequence Alignment of Peste des Petits Ruminants (PPR) Virus ChinaSX2020” documents isolation of a field strain of PPRV collected from a domestic milk goat. The authors describe how the isolate was obtained, and confirmed its etiology by different assays. 

  1. The manuscript needs some minor editing for proper English and style. 

Response: Thanks for your useful comments. The English has been improved.

  1. The abstract is disorganized and redundant. Identification of the virus before talking about RT-PCR?  

Response: Thanks for your comments. The abstract has been revised. It's our negligence. There's a mistake in the text, we used RT-PCR. It has been corrected in the whole manuscript R1 (Please see more in the revised manuscript R1).

  1. Line 157: the CT value does not bring anything to the results if there is no standard the reader can refer to.

Response: Thanks for your suggestive comments. The CT value was used to detect PPRV isolation. We then added the genomic copies of PPRV in lines 151-152 in the revised manuscript R1.

  1. Figure 1 A: the number of genomic copies are mentioned without mentioning the standard used. Is it per gram of tissues?

Response: Thanks for your useful suggestions. We detected the genomic copies of PPRV per gram of tissues in this study, and we added it in figure note in line 156 in the revised manuscript.

  1. Name of statistical test used for Figure 2C analysis is missing.

Response: Thanks. We added the statistical analysis for Figure 2C in lines 162-163 in the revised manuscript.

  1. it says that the data are normalized to an housekeeping gene. However, line 95 authors say stat analysis was performed using the delta delta method, which has nothing to do with how the data should be analyzed. Was it actually done?

Response: Thanks for your comments. Goat β-actin was used as a housekeeping gene. Data analysis was performed using the 2−∆∆CT relative quantification method (lines 95-96 and lines 98-100).

  1. Table 2: What was the origin of the sample giving the 100% reference value (negative control serum)? 

Response: In this study, we used the ELISA kit to detect the serum sample. Therefore, the negative control serum is in accordance with the instructions in line 126 in the revised manuscript.

  1. ChinaSX2020 is not plotted in the tree. There is a mislabeling. 

Response: Thanks. The ChinaSX2020 sequenced in this study is marked as a red dot in the tree (Figure 4) in line 192 in the revised manuscript.

Reviewer 2 Report

Please see the comments I made in the text of the pdf. 
All these recommendations contribute to the fact that I consider this paper to be publishable, provided that the requested modifications have been made, because it is not publishable as it stands from my point of view. 

Publishing on a single complete PPRV genome by manual method is quite rare now. The attractiveness of the paper must be particularly worked on the particular aspects of this strain in the set of the other strains already produced (GenBank). Its genome must be well verified as well as the sequences that enter the phylogenetic comparison. For theses reasons, I recommend that the analysis method be better adapted. On the other hand, PPR is now more and more documented on phylogenetic aspects and I recommend to the authors to refer to them, especially on the most recent publications because of the evolution of the situations on these very last years. I recommend the authors to take into consideration, for the suggestions made in the discussion, in particular on the vaccines to be used, to review their point of view in the light of the proven efficacy of these vaccines, of the success stories and of the regional and international ambitions of OIE, FAO regarding control and eradication strategy. 

Author Response

Thank you very much for your letter regarding our manuscript entitled “Analysis and Sequence Alignment of Peste Des Petits Ruminants Virus ChinaSX2020” (Manuscript Number: vetsci-1441706).

We also thank the anonymous reviewers for providing their comments and suggestions that are helpful for improving our manuscript. Based on their requests, we have carefully evaluated their comments and suggestions, responded point-by-point and revised the manuscript accordingly. Our point-by-point responses are listed below this letter.

The language has been also modified by native English speaking. We missed two authors when we submitted, and we have added them this time.

we have addressed all the comments from two reviewers. The files include:

  1. A point-by-point response to the reviewer comments (file: Responses to reviewers R1).
  2. The revised manuscript, all changes are highlighted in green (file: vetsci-1441706-R1).

We hope that the revised manuscript has satisfactorily addressed all the concerns raised by the reviewers.

Looking forward to hearing from you soon.

With kind regards,

Yours sincerely,

Lingxia Li & Youjun Shang

Reviewer #2

  1. Lacking: the number of full genome retrieved from GenBank, spanned years, country of origin (perhaps regions of China), host of origin (wildlife/domestic). If the evolutionary rates is foreseen, authors should use adapted method.

Bayesian time-scaled Maximum Clade Credibility should be used if authors like to speak about.

Response: Thanks for your comments. Bayesian method was used in our previous study about the genetic evolution analysis of PPRV, and that paper was under review now. While in this study, we just reported one strain of PPRV (ChinaSX2020).

  1. (A) The legend is incomplete, no reference to the standard used to quantify the copy numbers. In addition, this in not detailed in M&M. The different tissues are not cited in the legend. (B) We discover in this legend authors used RT-PCR based on primers targeting a N fragment. No detail is found in M&M.

Response: Thanks for your comments. The standard method used to quantify the copy numbers was constructed in our lab for many years. And this method has been patented by authors of this manuscript. In this study, PCR based on primers targeting H. While RT-PCR based on primers targeting a N fragment. The primers were from reference [18], and we added the reference in revised manuscript R1.

  1. There is no reference in M&M:

-of the use of the DAPI system.

-of the mean of measure of PPRV replication as said before (RT-PCR), and of type of data analysed.

-How the translation in copy number is made mRNA level (part C).

-It does not seem to be a correlation between the flurescence along time and mRNA level.

Response: Thanks for your suggestive comments. In order to identify PPRV, four different methods were used in this study (Fig. 2). (1) CPE was observed by microscope; (2) Indirect immunofluorescence (protein level, PPRV-N antibody); (3) qRT-PCR was used for PPRV replication (mRNA level, PPRV-H primers); (4) Virus titers of PPRV were determined by qRT-PCR (genomic copies). this method was established in our lab, and it has been patented.

  1. I don’t think this sentence should initiate the discussion. Why do authors talk only about wild-life, as the isolated strain was from a domestic animal? This sentence should be moved down to 196-197.

Response: Thanks. This sentence has been moved down.

  1. Authors should insist on quality control of the vaccines produced locally implemented by international bodies, rather than effectiveness of vaccine. As I said before, there are very effective vaccines, for example Nigeria75-1 vaccine which is produced worldwide, and also in China. The effectiveness of a vaccine relies on quality control of the production, to respect especially the minimum titre and the effectiveness of lyophilization. Delivery of the vaccine is important as well. If all these aspects are respected then vaccination is efficacious.

Authors should not make statements such as efficient vaccines are urgently needed, as this is an untruth and not only blame the vaccine. You have to better insist on the real causes of the circulation of the virus as you say below (206-211)

Response: Thanks very much. We have changed the unreasonable sentences and redescribed it. We have corrected all the mistakes in my article put forward by the reviewer in lines 207-211 in the revised manuscript.

  1. Authors should precise in this first sentence that they are talking about China.

Response: Thanks very much. We added talking about China precisely in the first sentence in lines 220-221 in the revised manuscript.

  1. This paragraph is not quite in link with the subject. I would suppress it.

Response: Thanks for your useful suggestions. we have deleted the paragraph which is not quite in link with the subject.

  1. Due to the rapid change in lineage geographic expansion, perhaps more recent publications can be given to highlight this aspect. Example for Africa: https://link.springer.com/article/10.1007/s00705-020-04732-1.

Response: Thanks for your useful suggestions. The closest references were also cited in lines 204-205 in the revised manuscript.

We have also handled all the comments raised by the reviewer #2 in the article (highlighted in green).

Round 2

Reviewer 1 Report

-Please revise your abstract. It is still very disorganized and confusing. The order in which the events are listed do not make sense. Edit for proper English as well.

-China SX2020 is still not listed in your figure 4. Please pay close attention.

-Use the word "laboratory" instead of "lab" in the manuscript. 

Author Response

This manuscript is a resubmission of an earlier submission. The following is a list of the peer review reports and author responses from that submission.

Round 1

Reviewer 1 Report

The manuscript “ Analysis and Sequence Alignment of Pest des Petits Ruminants (PPR) Virus ChinaSX2020” documents isolation of a field strain of PPRV collected from a domestic milk goat in Shanxi China in July 2020, named ChinaSX2020. The authors describes how the isolate was obtained, and confirmed its etiology by PCR amplifying the N gene, performing EM and IFA assays in-vitro in GTC cells, as well as by sequencing the full genome among other assays. 

Many flaws were found that make this manuscript not acceptable for publication.

Major comments:

  1. The manuscript needs some editing for proper English and style. Some statements are incomprehensible.
  2. The introduction and discussion need to be rewritten. It is disorganized, repetitive, and confusing. There is no clear goal identified. Legends need to be expanded.
  3. Lack of references in section 2, 3, and 4 to verify veracity of statements.
  4. There is no justification of why this particular goat from the Shaanxi county was chosen for analysis among the 573. Serological data should probably be shown first before the "laboratory analyses of PPRV"
  5. section 2. needs a statement regarding the statistical analysis that were performed on figure 2. Stars are shown in that figures but there are no explanation or description of the test used.
  6. section 2.3 does not say how the results were generated for that figure 2. Were the data normalized to an housekeeping gene prior plotting the relative amount of virus genome over time?
  7. section 2.3 authors refer to table 1 but this is not showing the primers. The corresponding table is missing. Primers used for nested PCR should be included.
  8. section 2.4. was the bELISA done with the N or the H viral protein? legend of table 1 says N but the description in section 2.4. says H. This assay requires the use of only one viral protein, not two. How were the percentage inhibition cut-off values decided? it is set at 60 and 40% but there is no explanation why. What was the origin of the sample giving the 100% reference value (negative control serum)? Also the wording "doubtful" is incorrect. It should be "inconclusive" instead. 
  9. Section 2.5. needs to be detailed.
  10. Section 3.2. ChinaSX2020 is actually not plotted in the tree. There is a mislabeling. No legend in the figure explain the meaning of the red symbols.

Minor comments:

  1. line 22: the virus is a member of the genus Morbillivirus in the family Paramyxoviridae
  2. line 26. this is an RNA-dependent RNA polymerase. 
  3. line 61 and 72. What do you mean?
  4. "Maybe" is not appropriate and is used multiple times in the manuscript. Statements should be rephrased.

Reviewer 2 Report

There is a contradiction. In line174 is written that ...therefore, efficient commercial vaccines are urgently needed [16]; in contrast in line 184 the authors says that ... efficacious vaccines are available...

Table 1 cited in line 88 is missing.

The sequencing method should be better explained.

Reviewer 3 Report

In presented manuscript “Analysis and Sequence Alignment of Pest Des Petits Ruminants (PPR) Virus ChinaSX202” authors perform Pest des Petit Ruminant Virus (PPRV) isolation from sample of dead goats (type of sample material is missing) on two different cell cultures (VERO and Goat tracheal epithelium cells (GTC)). Confirmation of the virus isolation was done by Electron microscopy, Indirect Immunofluorescence Assay, Reverse Transcriptase Polymerase Chain Reaction (RT-PCR) and full genome sequence by nested PCR assay (primers for sequencing are not provided). After full genome sequencing Phylogenetic three was constructed using Neighbor-Joining method with 1000 bootstrap repetitions and serological analysis from sheep samples was performed with blocking ELISA.

In general manuscript is hard to read and understand as English is very bad. Next to bad English, methodology is poorly explained together with result and discussion part. In general manuscript need to be rewrite and English language need to be improved. From the presented version is not possible to understand and follow work done as it not has fluency.

The chapter 3. results and discussion and 4. discussion and conclusions are not appropriate.

Line 164ff: switch to the discussion part of the manuscript.

In the phylogenetic tree no description of the red triangle was included.

The primers of the study are not listed in table 1.

What RNA extraction method was used?

What RT-qPCR was used? Description of the absolute quantification procedure is missing.

The serological survey had no connection to the other parts of the manuscript and should be removed. No sufficient information about the origin of the sera were presented (sera from what species, age of animals, maternal antibodies?).

I didn’t want to write every point what need to be changed as the manuscript is very bad written. If I did so review will be longer than manuscript.

My final decision is that manuscript need to be rejected.